# Neighbor Class Consistency on Unsupervised Domain Adaptation

## Abstract

Unsupervised domain adaptation (UDA) is to make a prediction for unlabeled data in a target domain with labeled data from the source domain available. Recent advances exploit entropy minimization and self-training to align the feature of two domains. However, as decision boundary is largely biased towards source data, class-wise pseudo labels generated by target predictions are usually very noisy, and trusting those noisy supervisions might potentially deteriorate the intrinsic target discriminative feature. Motivated by agglomerative clustering which assumes that features in the near neighborhood should be clustered together, we observe that target features from source pre-trained model are highly intrinsic discriminative and have a high probability of sharing the same label with their neighbors. Based on those observations, we propose a simple but effective method to impose Neighbor Class Consistency on target features to preserve and further strengthen the intrinsic discriminative nature of target data while regularizing the unified classifier less biased towards source data. We also introduce an entropy-based weighting scheme to help our framework more robust to the potential noisy neighbor supervision. We conduct ablation studies and extensive experiments on three UDA image classification benchmarks. Our method outperforms all existing UDA state-of-the-art.

## 1 Introduction

Recent advances in deep neural network have dominated many computer vision tasks, such as image recognition He et al. (2016), object detectionGirshick (2015), and semantic segmentationLong et al. (2015). However, collection and manual annotation need no trivial human effort, especially for vision tasks like semantic segmentation where dense annotations are required. Thanks to the growth of computer graphics field, it is now possible to leverage CNN to synthetic images with computer-generated annotations (Richter et al. (2016); Ros et al. (2016)), so unlimited amount of data with free annotation is available for training network in scale. However, directly applying the model trained on synthetic source data to unlabeled target data leads to performance degradation and Unsupervised Domain Adaptation (UDA) aims to tackle this domain shift problem.

A widespread of UDA methods were proposed to align the domain-invariant representations by simultaneously minimizing the source error and discrepancy(e.g. $\mathcal{H}$-divergence Ben-David et al. (2010); Hoffman et al. (2016) $\mathcal{H}\triangle\mathcal{H}$-divergenceBen-David et al. (2010)) between two domain such as the maximum mean discrepancy Tzeng et al. (2014), correlation distanceSun et al. (2016) and etc. Further, adversarial learning-based UDA Ganin & Lempitsky (2015); Tzeng et al. (2017); Radford et al. (2015); Hoffman et al. (2018); Tsai et al. (2018); Sankaranarayanan et al. (2018); Luo et al. (2019) methods aim to reduce this discrepancy between two domain by minimizing the adversarial loss. However, the major limitation of adversarial learning is that it only aligns the global feature distribution of two domains without considering the class labels. As the result, a small $\mathcal{H}\triangle\mathcal{H}$ distance does not guarantee the small error on ideal joint hypothesis on the features of two domains Liu et al. (2019).

To alleviate this issue, Entropy minimization (Grandvalet & Bengio (2005); Vu et al. (2019)) and Self-Training (Lee (2013); Zou et al. (2018)) are the two dominant methods to enforce the cluster assumption such that network can learn a discriminative feature space by pushing the decision boundary away from densely-distributed area. However, as decision boundary is largely biased to-

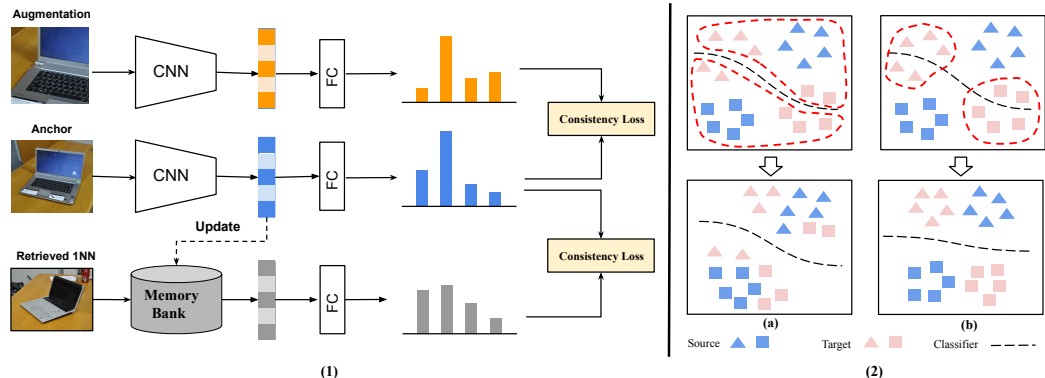

Figure 1: (1) Our Neighbor Class Consistency framework. (2) An overview of our approach: (a) Self-Training methods ignore the intrinsic target structure while aligning the features of two domains based on a biased classifier. They potentially deteriorate the intrinsic target clusters. (b) Our approach enforces the Neighbor Class Consistency on target features, therefore progressively strengthen the instrinsic discrimination of target features while regularizing the unified classifier less biased towards source data.

wards source data, trusting biased network predictions will push target features towards their nearest source class prototypes while deteriorating the intrinsic discriminative target structure as shown in Fig. 1(2.a).

Motivated by agglomerative clustering methods (Sarfraz et al. (2019)) which assume that features in the nearby region should be clustered together, we investigate target features from source pre-trained model and observe that they are intrinsically discriminative and have a very high possibility of sharing the same label with their neighbors as shown in Fig. 1(2.b). To utilize this high-quality pairwise neighbor supervision, we propose a simple and effective approach to impose Neighbor Class Consistency between target samples and their neighbors. To alleviate propagated errors from false neighbor supervision, we introduce an Entropy-based weighting scheme to emphasize more on the reliable pairwise neighbor supervision. Additionally, we categorize Self Class Consistency as a special case of our method where the nearest neighbor of a sample is its self-augmentation. Further, we explore feature representation learning based on the ranking relationship between self-augmentation and the first neighbor given an anchor. We enforce the features of anchors to be closer to their self-augmentation than their first neighbors.

In summary, our main contributions are shown as follows: (1) We revisit the source pre-trained model and observe the intrinsic discriminative nature of target features from source model. (2) Based on this observation, we propose Neighbor Class Consistency (NC) to utilize the high-quality pairwise neighbor pseudo supervision over noisy class-wise pseudo supervision from Self-Training methods. (3) We introduce an Entropy-based weighting scheme to help our framework be more robust to unreliable neighbor supervision. (4) We categorize Self Class Consistency as a special case of our framework and explore the first neighbor for feature representation learning. (5) We conduct extensive experiments on three UDA benchmarks datasets. NC outperforms all existing methods and achieves a new UDA state-of-the-art performance. Notably, we achieve 86.2% on challenging VisDA17 dataset.

## 2 RELATED WORK

**Discrepancy based domain adaptation** Following the theoretical upper bound proposed in Ben-David et al. (2007), existing methods have explored to align the feature representations of the source and target images by minimizing the distribution discrepancy. For example, Maximum Mean Discrepancy (MMD) Tzeng et al. (2014) is proposed to match the mean and covariance of source and target distributions. Alternatively, adversarial domain adaptation methods Ganin & Lempitsky (2015); Tzeng et al. (2017); Radford et al. (2015); Hoffman et al. (2018); Tsai et al. (2018); Sankaranarayanan et al. (2018); Luo et al. (2019) solve this domain discrepancy by training a domain-

invariant feature generator which produces the features to fool a discriminator that distinguishes the representations from source and target domains. However, since the domain discriminator aligns source and target features without considering the class labels, merely aligning the global marginal distribution of the features in the two domains fails to align the class-wise distribution.

**Clustering based domain adaptation** Entropy minimization (Grandvalet & Bengio (2005); Vu et al. (2019)). and self-training (Zou et al. (2018); Gu et al. (2020)) are two streams of approaches to realize the class-wise alignment across domains. However, minimizing the conditional entropy of target prediction based on source biased classifier will harm the intrinsic discriminative target structure. To this end, prototypical classifier and alignment Xie et al. (2018); Pan et al. (2019) have been explored to mitigate the noisy supervision from source biased classifier. However, the prototypes of target samples are still noisy estimated cluster centers and therefore target samples away from prototypes may still risk of being wrongly classified. Similar effort has been made by Tang et al. (2020) on learning the latent discriminative target structure with respect to learnable clusters via deep clustering framework Ghasedi Dizaji et al. (2017).

**Consistency Regularization** In semi-supervised setting, mainstream methods apply various consistency regularization on unlabeled data with different pre-defined positive counterpart. Among them, Tarvainen & Valpola (2017) impose consistency between predictions from student network and moving-average based teacher networks. Virtual Adversarial Training Miyato et al. (2018) tries to make network invariant to small adversarial permutations around the neighborhood of a sample while DTALee et al. (2019) enforces the target predictions from the networks with different choice of dropout mask to be consistent. In un/self-supervised setting, SimCLR (Chen et al. (2020)) and MoCo (He et al. (2020)) are the two prevalent approaches to conduct contrastive learning on the feature space among unlabeled data, its strong augmentation and other negative samples from either extremely large current batch or momentum updated memory banks. In comparison to those un/self-supervised methods, our method exploreS neighbor samples for consistency regularization. In cross-domain person re-identification, Zhong et al. (2019) also utilize the neighbor information on target data. To emphasize the difference, our method focuses on regularizing the classifier to be less biased toward source domain by applying target neighbor class consistency while they focus on the feature representation learning by enforcing neighbor feature invariance.

# 3 METHOD

## 3.1 PROBLEM DEFINITION

In unsupervised domain adaptation(UDA), source domain data is denoted as $\mathbb{D}_s = \{(\boldsymbol{x}_i^s, \boldsymbol{y}_i^s)|_{i=1}^{N_s}\}$ where $\boldsymbol{x}_i^s$ and $\boldsymbol{y}_i^s$ denote the $i$-th training sample and its label, $N_s$ is the number of source images. Target domain data is denoted as $\mathbb{D}_t = \{\boldsymbol{x}_i^t|_{i=1}^{N_t}\}$ where $N_t$ is the number of target images. The objective of UDA is to train a deep neural network $G(\cdot|\boldsymbol{\theta})$ which has access to the source data $(\boldsymbol{x}_i^s, \boldsymbol{y}_i^s)$ drawn from $\mathbb{D}_s$ and target data $\boldsymbol{x}_i^t$ drawn from $\mathbb{D}_t$ such that the model $G(\cdot|\boldsymbol{\theta})$ can generalize on target domain. Network $G(\cdot|\boldsymbol{\theta}) = C \circ F(\cdot|\boldsymbol{\theta})$ is comprised of a feature extractor $F(\cdot|\boldsymbol{\theta})$ and a classifier $C(\cdot|\boldsymbol{\theta})$ where $\boldsymbol{\theta}$ denotes network parameters.

## 3.2 REVISIT SUPERVISED PRE-TRAINING FOR SOURCE DOMAIN

In general, UDA pre-trains a network $G(\cdot|\boldsymbol{\theta})$ on source domain by standard cross entropy loss and then the network is transferred to inference the target data. The source supervised objective function is in the form of,

$$\mathcal{L}_{src}^s(\boldsymbol{\theta}) = \frac{1}{N_s} \sum_{i=1}^{N_s} \mathcal{L}_{ce}\left(C(F(\boldsymbol{x}_i^s|\boldsymbol{\theta})), \boldsymbol{y}_i^s\right). \tag{1}$$

However, the model trained on source data usually generalizes poorly on target data due to the domain shift between the joint distribution of two domains.

**Self-Training (ST) and Entropy Minimization (Ent)** methods are proposed to make the network be confident on its target predictions following the cluster assumption and thus improve the discrim-

| k-th Neighbor | k=1 | k=3 | k=5 |
|---|---|---|---|
| Acc | 96.4 | 92.3 | 86.4 |

Table 1: Pairwise class consistency evaluation on target features extracted from Source model with their k-th Nearest neighbors on Office-31 A→W.

inativeness of network. Their objective functions are calculated as

$$\mathcal{L}^t_{ST}(\boldsymbol{\theta}) = \frac{1}{N_t} \sum_{i=1}^{N_t} \mathcal{L}_{ce}\left(C(F(\boldsymbol{x}^t_i|\boldsymbol{\theta})), \tilde{\boldsymbol{y}}^t_i\right), \tag{2}$$

$$\mathcal{L}^t_{Ent}(\boldsymbol{\theta}) = \frac{1}{N_t} \sum_{i=1}^{N_t} \mathcal{H}\left(C(F(\boldsymbol{x}^t_i|\boldsymbol{\theta}))\right), \tag{3}$$

where $\tilde{\boldsymbol{y}}^t_i$ is the pseudo label for the $i$ target sample and $\mathcal{H}$ is entropy function.

Nonetheless, the pseudo labels of target samples are noisy as the network is biased towards source data. Trusting noisy label or prediction has high risk of misleading the training and propagate errors.

Inspired by agglomerative clustering methods which claim that features should be clustered with their near neighborhood, *we hypothesize that source domain pre-training can provide a feature space where target features are intrinsically discriminative and have high probability of sharing the same labels with their neighbors.*

To verify this, we apply k-Nearest Neighbor (kNN) search on target features extracted from source model on Offce-31 A→W. We evaluate the pairwise label consistent accuracy between target features and their k-th neighbors. Table 1 shows that target features from source model have very high accuracy of sharing the same label with their K-th neighbors. It demonstrates our hypothesis that target features from source model are locally discriminative.

This observation motivates us to leverage this relative "clean" pairwise pseudo supervision to help the network generalization on target domain over the noisy class-wise pseudo supervision.

### 3.3 Neighborhood Class Consistency

**1) Neighborhood Discovery.** First, we extract target features $\boldsymbol{z}^t = F(\boldsymbol{x}^t|\boldsymbol{\theta})$ from source pre-train model and save them into a target feature memory bank $\mathbb{V}_t = \{\boldsymbol{z}^t_i|_{i=1}^{N_t}\}$. Then we apply kNN on target features $\mathbb{V}_t$ to explore the neighborhood $\mathcal{N}_k(\boldsymbol{z}^t)$ which is defined as follows:

$$\mathcal{N}_k(\boldsymbol{z}^t_j) = \{\boldsymbol{z}^t_i | s(\boldsymbol{z}^t_i, \boldsymbol{z}^t_j) \text{ is top-k in } \mathbb{V}_t\}, \tag{4}$$

where $s$ is a similarity metric and $k$ is the number of neighbors.

Note that we update the memory bank $\mathbb{V}_t$ with the newest features in training and conduct kNN on $\mathbb{V}_t$ at every epoch.

**2) Vanilla Neighborhood Class Consistency (VNC).** Based on our motivation that target samples should share the same class label with their neighbors, we propose VNC to enforce the class assignment consistency between pairwise neighbors by mutual information (MI) maximization between their network predictions. Formally, we formulate the objective function of VNC in the form of:

$$\mathcal{L}^t_{VNC}(\boldsymbol{\theta}) = -\frac{1}{N_t} \sum_{i=1}^{N_t} \left(\frac{1}{k} \sum_{j \in \mathcal{N}_k(\boldsymbol{z}^t_i)} \mathcal{MI}\left(C(\boldsymbol{z}^t_i|\boldsymbol{\theta}), C(\boldsymbol{z}^t_j|\boldsymbol{\theta})\right)\right). \tag{5}$$

It is also worth noting that the memory bank serves as a look-up table to retrieve the target features and feed them into the classifier $C$ for computing the Neighborhood Class Consistency loss. As the size of $k$ can be potentially large, using memory bank can implicitly enlarge the batch size by $k$ times without introducing extra computational cost and time.

**3) Entropy-weighted Neighborhood Class Consistency (ENC).** Table 1 shows that the reliability of pairwise neighborhood supervision is decreased with the rise of neighborhood size $k$. In other words, there is a trade-off between large neighborhood diversity and reliability of neighbor supervision.

A straightforward idea is to treat each neighbor pairs differently in terms of loss weight such that hopefully positive neighbor pairs (Correct) will have higher loss weight than negative neighbor pairs (False). To achieve this, we propose a entropy-based weighting scheme (EW) to assign different loss weight on different neighbor pairs given an anchor. As the entropy is a measurement of sample prediction certainty, the less entropy is, more confident the prediction is and thus more weight we will assign on. Formally, we define the loss weight and objective of ENC as follows:

$$w(\boldsymbol{z}^t) = 1 - \frac{\mathcal{H}(C(\boldsymbol{z}^t|\boldsymbol{\theta}))}{\log M}, \tag{6}$$

$$\mathcal{L}_{ENC}^t(\boldsymbol{\theta}) = -\frac{1}{N_t} \sum_{i=1}^{N_t} \left( \frac{w(\boldsymbol{z}_j^t)}{k} \sum_{j \in \mathcal{N}_k(\boldsymbol{z}_i^t)} \mathcal{MI}\left(C(\boldsymbol{z}_i^t|\boldsymbol{\theta}), C(\boldsymbol{z}_j^t|\boldsymbol{\theta})\right) \right), \tag{7}$$

where $M$ is the number of class.

We claim that entropy-based weighting helps mitigate the problem of noisy pairwise supervision and it is more robust to the neighborhood size k as it will down-weight the neighbor pairs if the selected neighbor samples have high entropy value.

## 3.4 Self Class Consistency

In addition to neighborhood consistency, target samples should always share the same label with their self-augmentation and this pairwise supervision is 100% clean. Therefore, Self Class Consistency is a special case of NC where the nearest neighbor given an anchor is its augmentation. We adopt Cropping, grayscale and color distortion as our data augmentation following Chen et al. (2020) and enforce the self-consistency in terms of class assignment on target samples and their augmentations as follows,

$$\mathcal{L}_{SC}^t(\boldsymbol{\theta}) = -\frac{1}{N_t} \sum_{i=1}^{N_t} \mathcal{MI}\left(C(F(\boldsymbol{x}_i^t|\boldsymbol{\theta})), C(F(\tilde{\boldsymbol{x}}_i^t|\boldsymbol{\theta}))\right), \tag{8}$$

where $\tilde{\boldsymbol{x}}_i^t$ is the data augmentation of $i$-th target sample.

## 3.5 Feature Ranking between Self-Augmentation and First Neighbor

As the first neighbors given target anchors might be negative (do not share the same label with anchors) and the anchors' self-augmentations might be heavily distorted, enforcing neighbor class consistency alone might not guarantee that the anchors are closer to their self-augmented samples than their first neighbors in feature space. We claim that introducing this feature ranking regularization benefits the feature representation learning by ensuring the positive samples (self-augmentations) rank higher than the negative samples (the first neighbors) in feature space. When the first neighbors are positive, it can also enforce an inductive bias on model training to rank the positive samples with pre-defined variations (such as cropping, grayscale and color distortion) higher than the positive samples with unknown variations (the first neighbors).

To impose this feature ranking regularization, we adopt triplet loss as objective function as shown below,

$$\mathcal{L}_{tri}^t(\boldsymbol{\theta}) = \frac{1}{N_t} \sum_{i=1}^{N_t} \max\left(0, ||F(\boldsymbol{x}_i^t|\boldsymbol{\theta}) - F(\tilde{\boldsymbol{x}}_i^t|\boldsymbol{\theta})|| + m - ||F(\boldsymbol{x}_i^t|\boldsymbol{\theta}) - \mathcal{N}_1(F(\boldsymbol{x}_i^t|\boldsymbol{\theta}))||\right), \tag{9}$$

where $\mathcal{N}_1(F(\boldsymbol{x}_i^t|\boldsymbol{\theta}))$ is the feature of the first neighbor of target sample $i$ and $m$ is the margin.

### 3.6 TOTAL LOSS

The objective of proposed Neighbor Class Consistency is in the form of,

$$\mathcal{L}_{NC}^t(\boldsymbol{\theta}) = \mathcal{L}_{src}^s(\boldsymbol{\theta}) + \lambda(\mathcal{L}_{ENC}^t(\boldsymbol{\theta}) + \mathcal{L}_{SC}^t(\boldsymbol{\theta}) + \mathcal{L}_{tri}^t(\boldsymbol{\theta})). \qquad (10)$$

As pre-training on source data only might not provide the optimal initial feature space to explore high-quality neighborhood, we add target self-consistency loss with source supervised loss in pre-training stage and formulate it as:

$$\mathcal{L}_{SP}(\boldsymbol{\theta}) = \mathcal{L}_{src}^s(\boldsymbol{\theta}) + \lambda\mathcal{L}_{SC}^t(\boldsymbol{\theta}). \qquad (11)$$

## 4 EXPERIMENT

### 4.1 DATASETS

We conduct experiments on three widely used domain adaptation classification benchmarks: Office-31 Saenko et al. (2010), VisDA17 Peng et al. (2018) and ImageCLEF-DA [1].

**Office-31** is a commonly used dataset for unsupervised domain adaptation. It includes 4652 images of 31 classes from three domains: Amazon (A), Webcam (W) and DSLR (D). **ImageCLEF-DA** consists of 12 common classes shared by three public datasets (domains): Caltech-256 (C), ImageNet ILSVRC 2012 (I), and Pascal VOC 2012 (P). **VisDA17** is a large-scale dataset. It uses $152,409$ 2D synthetic images from 12 classes as the source training set and $55,400$ real images from MS-COCO Lin et al. (2014) as the target set. 12 object categories are shared by these two domains.

### 4.2 IMPLEMENTATION DETAILS

We follow the standard protocol of UDA( Ganin et al. (2016); Zhang et al. (2019b); Xu et al. (2019); Zou et al. (2019)) to use all labeled source samples and all unlabeled target samples as training data. The reported testing results are the average accuracy over three random repeats with center-crop images. We adopt ResNet-50 (He et al. (2016)) on Office-31 and ImageCLEF-DA while ResNet101 on VisDA17 dataset, fine-tuned from the ImageNet (Deng et al. (2009)) pre-trained model. We use Pytorch as implementation framework. We adopt Stochastic Gradient Descent (SGD) optimizer with learning rate of $1 \times 10^{-3}$, weight decay $5 \times 10^{-4}$, momentum 0.9 and batch size 32. We set $\lambda = 0.1$ (Eq 10) for Office-31 and ImageCLEF-DA while $\lambda = 0.5$ for VisDA17. We set $k = 1$ (Eq 7) to explore the most local neighbourhoods and triplet margin $m = 0.1$ in (Eq 9). The analysis on the impact of $k$ is at section 4.5.

### 4.3 ABLATION STUDY

In this section, we investigate the effectiveness of each components of NC in achieving the state-of-the art performance on Office-31. We name each components of NC as: (1) **Source Model**, which fine-tunes the base network on source labeled samples with Eq 1; (2) **Vanilla Neighbor Class Consistency (VNC)**, which denotes the training with non-weighted Neighbor Class consistency with Eq 5; (3) **Entropy-Weighted Neighbor Class Consistency (ENC)**, which denotes the training with Entropy-Weighted Neighbor Class consistency with Eq 7; (4) **Self Class Consistency(SC)**, which denotes the training with Self consistency with Eq 8; (5) **Feature Ranking (FR)**, which denotes the training with feature ranking relations between anchor, its self-augmentation and its first neighbor with Eq 9; (6) **Self-Supervised Pretraining (SP)**, which fine-tunes the base network with source labeled samples and SC with Eq 11. From the Tab 2, we observe that all proposed components play a contributing role to our final performance.

### 4.4 COMPARISONS WITH THE STATE OF THE ART

**Results on Office-31 and ImageCLEF-DA** based on ResNet-50 are shown in Table 3 and Table 4. We can observe that **NC-SP** outperforms all the existing methods and more importantly boosts the

---
[1]https://www.imageclef.org/2014/adaptation

| Method | A → W | D → W | W → D | A → D | D → A | W → A | Avg |
|---|---|---|---|---|---|---|---|
| Source Model | 68.4±0.2 | 96.7±0.1 | 99.3±0.1 | 68.9±0.2 | 62.5±0.3 | 60.7±0.3 | 76.1 |
| SC | 91.8 ±0.2 | 98.3±0.0 | 100.0±0.0 | 92.3±0.1 | 75.0±0.2 | 75.3±0.2 | 88.8 |
| VNC | 93.8 ±0.1 | 98.7±0.0 | 100.0±0.0 | 93.2±0.2 | 75.1±0.1 | 75.8±0.1 | 89.4 |
| ENC | 94.1±0.2 | 98.7±0.0 | 100.0±0.0 | 93.8 ±0.1 | 75.4±0.0 | 76.2 ±0.1 | 89.7 |
| ENC-SC | 94.2 ±0.1 | 98.7±0.0 | 100.0±0.0 | 95.8 ±0.2 | 75.3±0.1 | 76.7±0.3 | 90.1 |
| ENC-SC-SP | 95.8 ±0.1 | 98.7±0.0 | 100.0±0.0 | 95.4 ±0.1 | 76.9±0.1 | 77.0±0.1 | 90.6 |
| ENC-SC-FR (NC) | 95.7±0.2 | 98.7±0.0 | 100.0±0.0 | 95.2±0.1 | 77.5±0.1 | 76.9±0.0 | 90.6 |
| ENC-SC-FR-SP (NC-SP) | **96.4±0.1** | 98.7±0.0 | **100.0±0.0** | **95.8±0.3** | **77.6±0.2** | **77.4±0.2** | **91.0** |

Table 2: Ablation studies using Office-31 based on ResNet-50. Please refer to Section 4.3 on what each component represents.

| Method | A → W | D → W | W → D | A → D | D → A | W → A | Avg |
|---|---|---|---|---|---|---|---|
| ResNet-50 He et al. (2016) | 68.4±0.2 | 96.7±0.1 | 99.3±0.1 | 68.9±0.2 | 62.5±0.3 | 60.7±0.3 | 76.1 |
| DANN (Ganin et al. (2016)) | 82.0±0.4 | 96.9±0.2 | 99.1±0.1 | 79.7±0.4 | 68.2±0.4 | 67.4±0.5 | 82.2 |
| ADDA (Tzeng et al. (2017)) | 86.2±0.5 | 96.2±0.3 | 98.4±0.3 | 77.8±0.3 | 69.5±0.4 | 68.9±0.5 | 82.9 |
| ENT (Grandvalet & Bengio (2005)) | 89.0±0.1 | 99.0±0.1 | 100.0±0.0 | 86.3±0.3 | 67.5±0.2 | 63.0±0.1 | 84.1 |
| JAN (Long et al. (2017) ) | 85.4±0.3 | 97.4±0.2 | 99.8±0.2 | 84.7±0.3 | 68.6±0.3 | 70.0±0.4 | 84.3 |
| MRENT (Zou et al. (2019)) | 88.0±0.4 | 98.6±0.1 | 100.0±0.0 | 87.4±0.8 | 72.7±0.2 | 71.0±0.4 | 86.4 |
| SAFN+Ent (Xu et al. (2019)) | 90.1±0.8 | 98.6±0.2 | 99.8±0.0 | 90.7±0.5 | 73.0±0.2 | 70.2±0.3 | 87.1 |
| SymNets (Zhang et al. (2019a)) | 90.8±0.1 | 98.8±0.3 | 100.0±0.0 | 93.9±0.5 | 74.6±0.6 | 72.5±0.5 | 88.4 |
| MDD (Zhang et al. (2019b)) | 94.5±0.3 | 98.4±0.1 | 100.0±0.0 | 93.5±0.2 | 74.6±0.3 | 72.2±0.1 | 88.9 |
| CAN (Kang et al. (2019)) | 94.5±0.3 | 99.1±0.2 | 99.8±0.2 | 95.0±0.3 | **78.0±0.3** | 77.3±0.3 | 90.6 |
| SRDC (Tang et al. (2020)) | 95.7±0.2 | **99.2±0.1** | 100.0±0.0 | 95.8±0.2 | 76.7±0.3 | 77.1±0.1 | 90.8 |
| **NC-SP** | **96.4±0.1** | 98.7±0.0 | **100.0±0.0** | **95.8±0.3** | 77.6±0.2 | **77.4±0.2** | **91.0** |

Table 3: Experiment results on Office-31 classification using ResNet-50

performance substantially on difficult transfer tasks such as A→ W, D→ A and W→ A. Notably, comparing to state-of-the-art self-training (Zou et al. (2019)) and entropy minimization methods (Xu et al. (2019)) based on class-wise pseudo supervision, **NC-SP** outperforms them by 4.6% and 3.9% respectively, and it demonstrates that following the relatively clean neighborhood pseudo supervision can alleviate the error accumulation problem from self-training.

**Results on VisDA17** based on ResNet-101 are reported in Table 5. **NC-SP** achieves much better performance than all compared methods. Comparing to DTA (Lee et al. (2019)) which uses VAT (Miyato et al. (2018)) as self-consistency and Adversarial Dropout as model-consistency, we instead explore neighborhood consistency and demonstrate the effectiveness of **NC-SP** over other consistency regularization. Note: The results of existing methods in Table 3,4,5 refer to their respective papers.

### 4.5 Analysis

**Feature visualization.** We visualize the target feature embeddings of (a) source model, (b) entropy minimization, and (c) **NC** on Office-31 W→ A via t-SNE (Maaten & Hinton (2008)) in Fig.2. We can qualitatively observe that **NC** could learn more discriminative features than source model and entropy minimization as it preserves the intrinsic target features structure.

**Nearest neighbors visualization.** We visualize the top 3 nearest neighbors given an anchor based on the target features from source model on Office-31 A → W in Fig.3(a). We investigate both success and failure cases to get extra insights into our method. For the first two row, features from source model could retrieve the correct neighbors for the mobile phone and backpack with certain level of appearance and pose variations. However, at the last row where the pose of that laptop sample is unusual, our method might fail in those cases. The proposed entropy weighting on neighbor class consistency is to alleviate the misleading learning by false neighborhood supervision.

**Impact of neighbourhood size $k$.** Neighbourhood size $k$ is an important parameter as it controls the amount of pairwise neighborhood supervision. However, there is a trade-off between increasing the neighbor diversity and increasing the risk of adding false neighborhood supervision. Empirically, the optimal $k$ is associated with the size of target dataset and the number of class. We evaluate $k$ from $\{1, 3, 5\}$ for both Vanilla NC and Entropy-weighted NC on Office-31 A → W as shown in Fig 3(b). We observe that the performance decreases when increasing the $k$ for VNC while ENC is more

| Method | I→P | P→I | I→C | C→I | C→P | P→C | Avg |
|---|---|---|---|---|---|---|---|
| ResNet-50 (He et al. (2016)) | 74.8 | 83.9 | 91.5 | 78.0 | 65.5 | 91.2 | 80.7 |
| DANN (Ganin et al. (2016)) | 75.0 | 86.0 | 96.2 | 87.0 | 74.3 | 91.5 | 85.0 |
| JAN (Long et al. (2017)) | 76.8 | 88.0 | 94.7 | 89.5 | 74.2 | 91.7 | 85.8 |
| CDAN (Long et al. (2018)) | 76.7 | 90.6 | 97.0 | 90.5 | 74.5 | 93.5 | 87.1 |
| SAFN+Ent (Xu et al. (2019)) | 78.0 | 91.7 | 96.2 | 91.1 | 77.0 | 94.7 | 88.1 |
| SymNets (Zhang et al. (2019a)) | 80.2 | 93.6 | 97.0 | 93.4 | 78.7 | 96.4 | 89.9 |
| CAN (Kang et al. (2019)) | 78.5 | 93.0 | 97.3 | 91.0 | 77.2 | 97.0 | 89.0 |
| A$^2$LP+CAN (Zhang et al. (2020)) | 79.8 | 94.3 | 97.7 | 93.0 | **79.9** | 96.9 | 90.3 |
| NC-SP | **80.9** | **95.0** | **97.9** | **94.2** | 79.8 | **97.5** | **90.9** |

Table 4: Experiment results on ImageCLEF-DA classification using ResNet-50

| Method | Aero | Bike | Bus | Car | Horse | Knife | Motor | Person | Plant | Skateboard | Train | Truck | Mean |
|---|---|---|---|---|---|---|---|---|---|---|---|---|---|
| Source (Saito et al. (2018a)) | 55.1 | 53.3 | 61.9 | 59.1 | 80.6 | 17.9 | 79.7 | 31.2 | 81.0 | 26.5 | 73.5 | 8.5 | 52.4 |
| DANN( Ganin et al. (2016)) | 81.9 | 77.7 | 82.8 | 44.3 | 81.2 | 29.5 | 65.1 | 28.6 | 51.9 | 54.6 | 82.8 | 7.8 | 57.4 |
| MCD (Saito et al. (2018b) ) | 87.0 | 60.9 | 83.7 | 64.0 | 88.9 | 79.6 | 84.7 | 76.9 | 88.6 | 40.3 | 83.0 | 25.8 | 71.9 |
| ADR (Saito et al. (2018a)) | 87.8 | 79.5 | 83.7 | 65.3 | 92.3 | 61.8 | 88.9 | 73.2 | 87.8 | 60.0 | 85.5 | 32.3 | 74.8 |
| SAFN (Xu et al. (2019)) | 93.6 | 61.3 | 84.1 | 70.6 | 94.1 | 79.0 | 91.8 | 79.6 | 89.9 | 55.6 | 89.0 | 24.4 | 76.1 |
| MRKLD+LRENT (Zou et al. (2019)) | 88.0 | 79.2 | 61.0 | 60.0 | 87.5 | 81.4 | 86.3 | 78.8 | 85.6 | 86.6 | 73.9 | **68.8** | 78.1 |
| DTA (Lee et al. (2019)) | 93.7 | 82.2 | 85.6 | 83.8 | 93.0 | 81.0 | 90.7 | 82.1 | **95.1** | 78.1 | 86.4 | 32.1 | 81.5 |
| RWOT (Xu et al. (2020)) | 95.1 | 80.3 | 83.7 | **90.0** | 92.4 | 68.0 | **92.5** | **82.2** | 87.9 | 78.4 | **90.4** | 68.2 | 84.0 |
| NC-SP | **97.1** | **88.5** | **90.0** | 65.2 | **96.7** | **92.9** | 90.1 | 81.5 | 94.6 | **89.5** | 89.0 | 58.8 | **86.2** |

Table 5: Experimental results on VisDA17 classification using ResNet-101.

robust to $k$. This is consistent with the intuition of Entropy-weighting which aims to alleviate the problem of false neighborhood supervision. For simplicity, we use $k=1$ through all the experiments while increasing $k$ definitely has lots of potential to boost the performance further especially when applying Entropy weighting scheme.

**How does entropy-based weighting looks like?** In order to prove our hypothesis that negative neighbor pairs will have lower consistency loss weight, we compute the entropy weights for all the neighbor pairs ($k=1$) on Office-31 A → W and W → A, and take average over entropy weights on positive pairs and negative pairs respectively. From Fig 3(c), we can see that positive neighbor pairs have much higher consistency loss weight than negative pairs by 20% on average. *It demonstrates our hypothesis that Entropy-weighted NC could alleviate the false neighbor supervision via assigning less loss weight to them.*

**Hyper-parameter sensitivity.** We conduct the hyper-parameter sensitivity analysis on the $\lambda$ in Eqn. 10 and the triplet margin $m$ in Eqn. 9 on Office31 A → W. As Fig. 5 shows, the performance of VNC decreases with the increase of $\lambda$ while the performance of NC is inversely proportional to the triplet margin $m$. To conclude, our method is robust to the regional changes of hyper-parameters. The reason that large margin results in a performance drop might be that there exists a portion of anchors whose first neighbors are positive and semantically more similar to the anchors than their self-augmentations. Setting the margin to a very large value will emphasize the inductive bias too much on the pre-defined variations and thus harm our model to align those positive neighbors with unknown variations to the anchors. By setting a small margin, our method could ensure the self-augmentations are more close to their anchors when the first neighbors of anchors are negative. At the same time, we can ensure that the self-augmentations are not too far away from the first neighbors when they are positive.

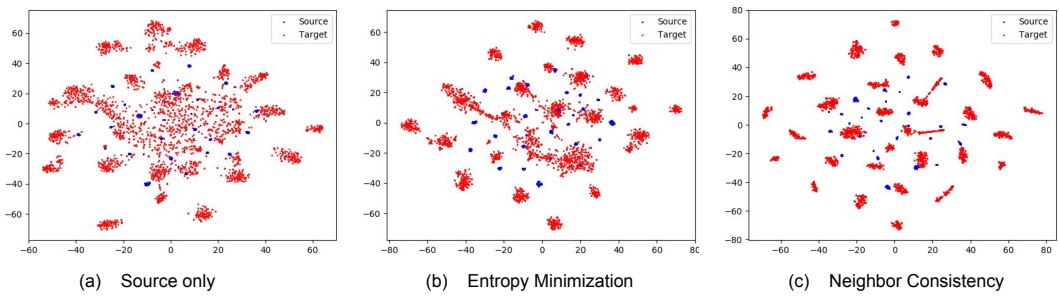

(a)  Source only          (b)  Entropy Minimization          (c)  Neighbor Consistency

Figure 2: The t-SNE visualization of target feature embedding (red) on Office-31 $W \to A$.

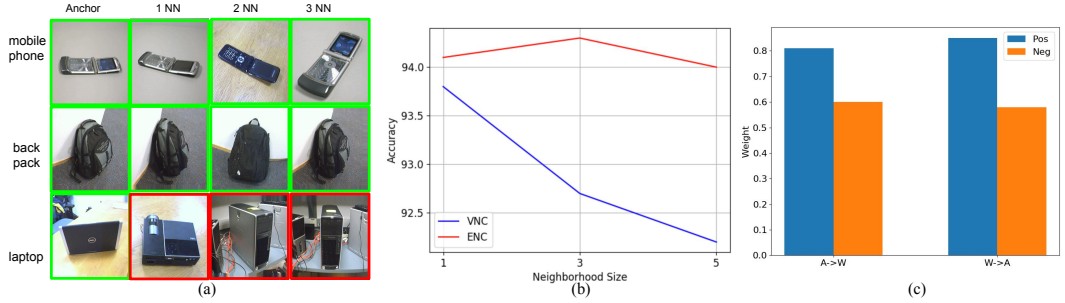

Figure 3: Analysis: (a) Top 3 nearest neighbors given an anchor based on source model on Office-31. (Green: positive sample; Red: negative sample.) (b) Impact of the neighbourhood size K on Office-31 A → W. (c) Entropy-based weighting for positive and negative neighbor pairs on Office-31 A → W, W → A.

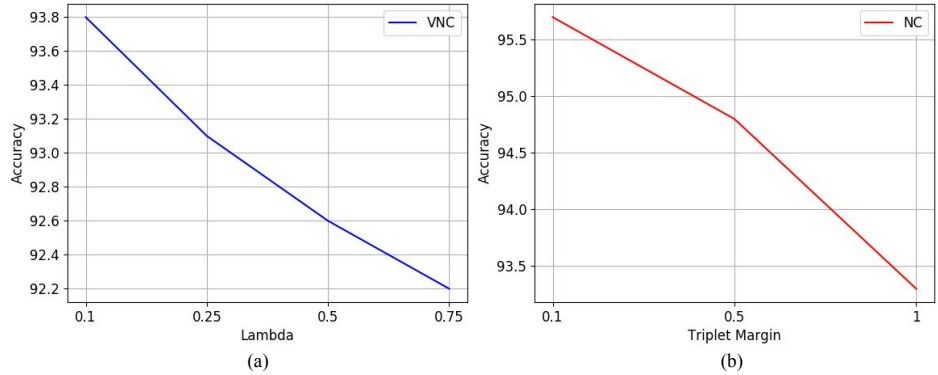

Figure 4: Hyper-parameter sensitivity analysis: (a) Sensitivity of $\lambda$ on Office-31 A → W. (b) Sensitivity of $m$ on Office-31 A → W.

## 5 CONCLUSION

In this work, we propose a simple but effective appoarch, named Neighbor Class Consistency based on the observation that target features extracted from source pre-trained model are high intrinsically discriminative. We introduce an entropy-based weighting scheme to improve the robustness of our framework to potential noisy neighbor supervision. We also incorporate self-supervision and impose metric learning based on feature ranking relationship. We conduct solid ablation studies to prove each proposed components contributing to the performance and extensive experiments on three UDA image classification benchmarks. Our method outperforms all existing UDA state-of-the-art.

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

## A  APPENDIX

### A.1  ADDITIONAL HYPER-PARAMETER SENSITIVITY ANALYSIS

To select the best neighborhood size $k$ and analyze the influence of $k$ on the performance, we evaluate $k$ from $\{1, 3, 5\}$ for all the tasks on Office31 and report the performance in Fig. 5. We observe that our method (NC) achieves the best performance when $k = 1$ while the performance does not monotonically decrease with the increase of neighborhood size.

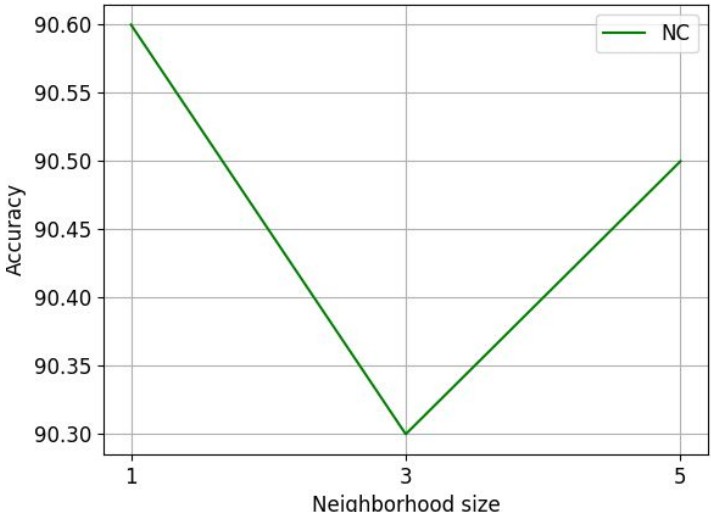

Figure 5: Hyper-parameter sensitivity of neighborhood size $k$ on Office-31 for all tasks.

## A.2 COMPARISON TO THE RELATED WORK FROM OTHER DOMAIN

To differentiate our method to the related work from other domains such as self-supervised learning (Chen et al. (2020)), knowledge distillation (Yun et al. (2020)) and person re-identification tasks (Zhong et al. (2019)), we implement their idea in unsupervised domain adaptation setting on Office31 A $\rightarrow$ W in table 6. Specifically, 1) for self-supervised learning, we utilize the self consistency loss in Eqn. 8 only as comparison and term it as **SC (ours)**. Though our objective function is different, we adopt the same data augmentation rule from from (Chen et al. (2020)). 2) For knowledge distillation which is a supervised setting, we replace our objective function (Mutual information maximization) in our ENC to the KL divergence objectives from (Yun et al. (2020)) to compute the class consistency loss and term it as **CS-KD**. 3) For person re-identification, we refer to the official github repository of (Zhong et al. (2019)) which introduces the neighbors invariance in feature space and implement it in UDA setting. We term it as **ECN**.

As table 6 shows, our ENC outperforms other baselines by a significant margin. It demonstrates that 1) the neighbor class consistency contributes more to the domain alignment than self consistency loss. 2) Our consistency loss based on mutual information maximization is better than KL divergence in terms of enforcing class prediction consistency. 3) Our neighbor class consistency loss is more effective than enforcing neighbor feature invariance. We think the features of target neighbors are naturally close to each other in feature space but the source biased classifier fails to provide consistent class predictions between the target neighbor. Therefore, in our point of view, providing regularization on the classifier via our neighbor class consistency loss is more important than enforcing neighbor feature invariance in UDA problem.

| Method | A $\rightarrow$ W |
|---|---|
| ECN (Zhong et al. (2019)) | 87.9 |
| CS-KD (Yun et al. (2020)) | 84.5 |
| SC (ours) | 91.8 |
| ENC (ours) | 94.1 |

Table 6: Comparison to the related works on Office-31 A$\rightarrow$W where ECN explores neighbor feature invariance and CS-KD utilizes the class-wise prediction regularization.

## A.3 Adding Results for ENC on ImageCLEF-DA and VisDA17

To show the effectiveness of our method with ENC only, we report the results on ImageCLEF-DA and VisDA17 in table 7 and 8. We could observe that our method with ENC only achieves the third best performance (exclude NC-SP) in Office-31, top 1 performance in ImageCLEF-DA and the second best performance in VisDA17. Compared to the self-training based methods (ENT, MRENT, SAFN+Ent), our ENC outperforms them by a large margin without bells and whistles.

| Method | I→P | P→I | I→C | C→I | C→P | P→C | Avg |
|---|---|---|---|---|---|---|---|
| ResNet-50 (He et al. (2016)) | 74.8 | 83.9 | 91.5 | 78.0 | 65.5 | 91.2 | 80.7 |
| DANN (Ganin et al. (2016)) | 75.0 | 86.0 | 96.2 | 87.0 | 74.3 | 91.5 | 85.0 |
| JAN (Long et al. (2017)) | 76.8 | 88.0 | 94.7 | 89.5 | 74.2 | 91.7 | 85.8 |
| CDAN (Long et al. (2018)) | 76.7 | 90.6 | 97.0 | 90.5 | 74.5 | 93.5 | 87.1 |
| SAFN+Ent (Xu et al. (2019)) | 78.0 | 91.7 | 96.2 | 91.1 | 77.0 | 94.7 | 88.1 |
| SymNets (Zhang et al. (2019a)) | 80.2 | 93.6 | 97.0 | 93.4 | 78.7 | 96.4 | 89.9 |
| A$^2$LP+CAN (Zhang et al. (2020)) | 79.8 | 94.3 | 97.7 | 93.0 | **79.9** | 96.9 | 90.3 |
| ENC | 79.6 | 95.0 | 97.5 | 94.1 | 79.2 | 96.7 | 90.3 |
| NC-SP | **80.9** | **95.0** | **97.9** | **94.2** | 79.8 | **97.5** | **90.9** |

Table 7: Experiment results on ImageCLEF-DA classification using ResNet-50

| Method | Aero | Bike | Bus | Car | Horse | Knife | Motor | Person | Plant | Skateboard | Train | Truck | Mean |
|---|---|---|---|---|---|---|---|---|---|---|---|---|---|
| Source (Saito et al. (2018a)) | 55.1 | 53.3 | 61.9 | 59.1 | 80.6 | 17.9 | 79.7 | 31.2 | 81.0 | 26.5 | 73.5 | 8.5 | 52.4 |
| DANN( Ganin et al. (2016)) | 81.9 | 77.7 | 82.8 | 44.3 | 81.2 | 29.5 | 65.1 | 28.6 | 51.9 | 54.6 | 82.8 | 7.8 | 57.4 |
| MCD (Saito et al. (2018b) ) | 87.0 | 60.9 | 83.7 | 64.0 | 88.9 | 79.6 | 84.7 | 76.9 | 88.6 | 40.3 | 83.0 | 25.8 | 71.9 |
| ADR (Saito et al. (2018a)) | 87.8 | 79.5 | 83.7 | 65.3 | 92.3 | 61.8 | 88.9 | 73.2 | 87.8 | 60.0 | 85.5 | 32.3 | 74.8 |
| SAFN (Xu et al. (2019)) | 93.6 | 61.3 | 84.1 | 70.6 | 94.1 | 79.0 | 91.8 | 79.6 | 89.9 | 55.6 | 89.0 | 24.4 | 76.1 |
| MRKLD+LRENT (Zou et al. (2019)) | 88.0 | 79.2 | 61.0 | 60.0 | 87.5 | 81.4 | 86.3 | 78.8 | 85.6 | 86.6 | 73.9 | **68.8** | 78.1 |
| DTA (Lee et al. (2019)) | 93.7 | 82.2 | 85.6 | 83.8 | 93.0 | 81.0 | 90.7 | 82.1 | **95.1** | 78.1 | 86.4 | 32.1 | 81.5 |
| RWOT (Xu et al. (2020)) | 95.1 | 80.3 | 83.7 | **90.0** | 92.4 | 68.0 | **92.5** | **82.2** | 87.9 | 78.4 | **90.4** | 68.2 | 84.0 |
| ENC | 96.9 | 89.6 | 85.3 | 70.8 | 96.4 | 87.8 | 92.4 | 76.7 | 91.6 | 84.1 | 87.8 | 41.8 | 83.5 |
| NC-SP | **97.1** | **88.5** | **90.0** | 65.2 | **96.7** | **92.9** | 90.1 | 81.5 | 94.6 | **89.5** | 89.0 | 58.8 | **86.2** |

Table 8: Experimental results on VisDA17 classification using ResNet-101.

## A.4 Mutual Information objective function

We define the class prediction of target sample i as $P(\boldsymbol{z}_i^t) = C(\boldsymbol{z}_i^t|\boldsymbol{\theta})$ and its neighbor sample j as $P(\boldsymbol{z}_j^t) = C(\boldsymbol{z}_j^t|\boldsymbol{\theta})$.

The objective of mutual information can be formulated as:

$$\mathcal{MI}\left(P(\boldsymbol{z}_i^t), P(\boldsymbol{z}_j^t)\right) = P(\boldsymbol{z}_i^t, \boldsymbol{z}_j^t) \log \frac{P(\boldsymbol{z}_i^t, \boldsymbol{z}_j^t)}{P(\boldsymbol{z}_i^t)P(\boldsymbol{z}_j^t)}, \tag{12}$$

where

$$P(\boldsymbol{z}_i^t, \boldsymbol{z}_j^t) = \frac{1}{N_t} \sum_{i=1}^{N_t} \frac{(P(\boldsymbol{z}_i^t)P(\boldsymbol{z}_j^t)^T + P(\boldsymbol{z}_j^t)P(\boldsymbol{z}_i^t)^T)}{2}, \tag{13}$$

