# OpenReview forum: "Neighbor Class Consistency on Unsupervised Domain Adaptation"
_ICLR.cc/2021/Conference — Reject_

### Official Review · AnonReviewer1 · 2020-10-26
**Official Blind Review #1**

**Rating:** 4
**Confidence:** 5

**Review:**

This paper proposed neighbor class consistency regularization together with an entropy-based weighting factor to tackle the problem of unsupervised domain adaptation. Another self class consistency regularization was further introduced to help training. The difference between "neighbor class" and "self class" is the positive pair selection, where "neighbor class" uses k-nearest neighbors as positive pairs, and "self class" uses an augmented version of the anchor itself.

The paper is generally well-written and easy to follow. Sufficient experiments are conducted. However, I think the quality of the paper is not publishable now due to the following weaknesses.

Major weaknesses:

1. The core contribution of this paper is the class consistency loss (Eq. (5)&(7)) between the anchor and its neighbors, however, similar ideas have been investigated in existing works, e.g. [A, B]. But unfortunately, this paper even did not include them in the related works.



2. I think the work lacks novelty. The memory bank in Sec 3.3 (1) is similar to [C, D] and the data augmentation rules in Sec 3.4 are similar to [E]. The consistency loss is popular in knowledge distillation tasks. In my point of view, the paper heavily borrowed techniques from existing works.  The author should at least acknowledge the relations between the paper and these works.



3. It's somewhat counter-intuitive that the paper required the similarity between the anchor and its augmented version to be closer than the similarity between the anchor and its nearest neighbor (Eq. (9)). A situation may exist: the augmentation is very heavy while the nearest neighbor shows very similar visual features with the anchor.
The author has shown the effectiveness of Eq. (9) in Table 2, namely "FR". However, it's not convincing enough, I recommend to evaluate "ENC + SC + SP (NC-SP)", i.e. removing "FR" from the whole model.


Minor weaknesses:

4. The paper claimed its state-of-the-art performances on various benchmarks. But it missed some previous methods, e.g. [F]. The paper achieved even inferior performance than [F] on the VisDA17 benchmark, i.e. 86.2 in this paper v.s. 87.2 in [F].

5. As I mentioned in weakness-3, it's better to use "minus" rather than "plus" in ablation studies.

6. More hyper-parameter analysis would help improve paper quality. For example, as described in Sec 4.2, the author adopted different $\lambda$ values for different datasets. It's better to show how much does the value of $\lambda$ impact the final performance.



[A] Regularizing Class-wise Predictions via Self-knowledge Distillation. CVPR 2020.

[B] Invariance Matters: Exemplar Memory for Domain Adaptive Person Re-identification. CVPR 2019.

[C] Unsupervised Feature Learning via Non-Parametric Instance Discrimination. CVPR 2018.

[D] Momentum Contrast for Unsupervised Visual Representation Learning. CVPR 2020.

[E] A Simple Framework for Contrastive Learning of Visual Representations. ICML 2020.

[F] Contrastive Adaptation Network for Unsupervised Domain Adaptation. CVPR 2019.


=========================================================================================================

Update:

Thanks for the authors' response. But unfortunately, the most significant concern has not been addressed well, which is regarding of the paper's novelty. I thought the contributions of this work are incremental, and the authors' rebuttal did not convince me well. Btw, the authors claimed that [A] used KL-divergence while their work used mutual information maximization, however, minimizing KL-divergence is actually one kind of mutual information maximization. What's more, the authors even did not clearly point out which kind of mutual information maximization they used in either the manuscript or the rebuttal. Thanks again for the authors' efforts, but I choose to maintain my original score.

---

> ### Author Response · Authors · 2020-11-25
> **Response to Q1**
>
> `R1-Q1`:  The core contribution of this paper is the class consistency loss (Eq. (5)&(7)) between the anchor and its neighbors, however, similar ideas have been investigated in existing works, e.g. [A, B]. But unfortunately, this paper even did not include them in the related works.
>
> `R1-A1`: Thanks for pointing out those insightful papers. We will definitely cite them into our final version and explain the relations and differences between them.
>
> **Compared to [B]**:,  we agree that we share high-level similarity with [B] in terms of exploring neighbors information. But we would like to kindly argue the essential difference that  **they focus on enforcing the neighbor features invariance while we focus on the neighbor class prediction consistency**.  We also introduce a new entropy-based weighting scheme and Feature ranking constraint where [B] hasn't explored.  We will explain those core difference in detail and demonstrate that those difference will result in a large distinction in terms of performance as follows:
>
> 1) **The problem setting, and methodology focus are different**:
> (a) **Unsupervised cross-domain person Re-ID**: The key difference between [B] and our method lies in the different settings between unsupervised domain adaptation (UDA) and Unsupervised cross-domain person RID (UDA RID). In UDA RID, the source and target data do not share any class label and the number of classes for target domain is unknown. As the source classifier cannot be applied to target data, [B] follows [C] to utilize an instance-level classifier for target data. Specifically, they store each target feature into the memory bank and use the memory bank as the instance-level classifier to make the target neighbors be close in the feature space in a multi-label learning manner. (Please refer to the Github repo of B for more details: https://github.com/zhunzhong07/ECN/blob/master/reid/loss/invariance.py). So, [B] essentially enforces the neighbor features to be close.
> (b) **Unsupervised domain adaptation**:In comparison, we can use the classifier trained on source to make class predictions for target data as the source and target data share the label space in UDA setting. We think the features of target neighbors are naturally close to each other in feature space but the source biased classifier fails to provide consistent class predictions between them. Therefore, in our point of view,  providing regularization on the classifier via our neighbor class consistency loss is more important than enforcing neighbor feature invariance from [B] in UDA problem. Further, we claim our main contribution is on providing a new regularization method on the shared classifier to make it less biased toward source data. Even though we share conceptual similarity with [B] by using neighbor information, our method is essentially different from [B] in terms of the functionality of methods.
>
>
> 2) **Our method significantly outperforms [B] in UDA setting**: To quantitatively demonstrate the difference between neighbor features invariance and neighbor class prediction consistency, we follow the official github repo of [B] at https://github.com/zhunzhong07/ECN/blob/master/reid/loss/invariance.py to implement the neighbor features invariance idea in UDA setting on Office31 dataset A->W task.  From the table 6 in appendix, we can see that our method outperforms them (ECN) by a significant margin (6.2%). It demonstrates that providing regularization on the source biased classifier via neighbor class consistency is more effective than enforcing neighbor features invariance which it is already naturally preserved to some degree.
> | Method | A->W |
> |-|-|
> | ECN [B]| 87.9|
> | ENC (ours) | 94.1 |
>
> 3)**The loss design and functionality of the memory bank are different**: From the loss function perspective, our method is also different from [B]. We enforce the neighbor class consistency via maximizing the mutual information between class predictions of anchor and neighbor. In comparison, [B] utilizes the standard cross-entropy with multi-label ground truth.
> Also, though we both utilize the memory bank, the functionality of the memory bank is different too. [B] utilize memory bank as instance-level classifier while we just use it as a lookup table to retrieve the target neighbor features for computing loss and save the computational time for the inference of those neighbor samples.
>
> (To be continued on the next page.)
>
> [A] Regularizing Class-wise Predictions via Self-knowledge Distillation. CVPR 2020.
>
> [B] Invariance Matters: Exemplar Memory for Domain Adaptive Person Re-identification. CVPR 2019.
>
> [C] Unsupervised Feature Learning via Non-Parametric Instance Discrimination. CVPR 2018.
>
> [D] Momentum Contrast for Unsupervised Visual Representation Learning. CVPR 2020.
>
> [E] A Simple Framework for Contrastive Learning of Visual Representations. ICML 2020.
>
> [F] Contrastive Adaptation Network for Unsupervised Domain Adaptation. CVPR 2019.

---

> ### Author Response · Authors · 2020-11-25
> **Response to Q1 and Q2.**
>
> `R1-Q1`:   ...similar ideas have been investigated in existing works, e.g. [A, B].
>
> `R1-A1`:  **Compared to [A]**,  even though we both explore the class-wise prediction consistency, we want to kindly specify the two major differences as follows:
>
> 1) **Problem setting and the choice of positive samples are different**: [A] choose the two different samples with the same label and enforce the class-wise prediction consistency between these two samples. So, [A] is essentially supervised learning problem. In contrast, we do not know the label for target data in the UDA setting. Instead, we choose a target sample with its self-augmentation and neighbors to be class consistent which is novel in the UDA domain.
>
> 2) **Objective function is different while ours is a better objective to enforce class consistency**:  Additionally, [A] utilizes KL divergence as an objective function for class-wise consistency while we use mutual information maximization as objectives. To demonstrate the effectiveness of our loss function with mutual information (MI), we replace the MI loss term with the KL divergence in our ENC experiment for comparison and term it as CS-KD. As table 6 from the appendix shows, using MI as an objective function achieves much better performance than KL divergence by  9.6%.
>
> | Method | A->W |
> |-|-|
> | CS-KD [A] | 84.5 |
> | ENC (ours) | 94.1 |
>
> `R1-Q2`: I think the work lacks novelty. The memory bank in Sec 3.3 (1) is similar to [C, D] and the data augmentation rules in Sec 3.4 are similar to [E]. The paper heavily borrowed techniques from existing works. The author should at least acknowledge the relations between the paper and these works.
>
> `R1-A2`: Thanks for your valuable comments. We would like to kindly argue the difference to [C, D] in terms of the memory bank and data augmentation. Further, we want to emphasize our novelty is non-trivial in the UDA domain.
> 1) **Functionality of memory bank is different**:As we reply in A1, [B,C,D] all utilize memory bank as an instance-level classifier which consists of the features of unlabeled samples. In contrast, we are solving a close-set classification problem in UDA where the source classifier is applicable. Therefore, we just utilize the memory bank for computational saving purposes. Specifically, we use it as a lookup table to retrieve the neighbor features given an anchor, feed them into the classifier, and computing the consistency loss.
>
> 2) **The objective function of utilizing data augmentation is different**: We utilize Mutual information as an objective function to enforce class prediction consistency between anchors and self-augmentation while [E] utilizes the cross-entropy as an objective to enforce feature representation between self-augmentations being invariant. Further, we emphasize that our major contribution is on neighbor class consistency where self-augmentation is a special case. To show the performance self class consistency only, we add more ablation studies in table 2 for SC only. We can see that SC alone can achieve a decent performance compared to source only models but is inferior to Vanilla VNC by 0.6%. Therefore, our major components for boosting the performance are still neighbor class consistency.
>
> 3) **Summary of our novelty**:  Motivated by the self-training methods which generally suffer from the class-wise noisy pseudo labels, we conduct the first attempt to leverage the more clean pairwise pseudo label in the closed-set UDA problem.  To differentiate from the related work where the classifier is missing, our major contribution is on providing a regularization on the shared classifier to make it less biased toward source data by introducing neighbor class consistency via Mutual information maximization. We demonstrate the neighbor class consistency is a more important factor than neighbor feature invariance in Table 6. We also introduce several important modules such as ENC to alleviate wrong supervision, FR to enforce the feature ranking relations, SP to initialize a good feature representation.
> As you notice that consistency loss is popular and important in both self-distillation, semi/unsupervised learning problems, the majority of previous works [A, C,D,E,G,H] are either utilizing different forms of self-supervision (data augmentation) in semi/unsupervised setting or utilizing label to explore positive samples given anchors in self-distillation (supervised) setting. In contrast, our method is novel in terms of leveraging neighbor pairwise supervision in an unsupervised setting.
>
>
> [G] Takeru Miyato, Shin-ichi Maeda, Masanori Koyama, and Shin Ishii.  Virtual adversarial training: a regularization method for supervised and semi-supervised learning. TPAMI2018
>
> [H] Seungmin  Lee,  Dongwan  Kim,  Namil  Kim,  and  Seong-Gyun  Jeong.   Drop  to  adapt:  Learning discriminative features for unsupervised domain adaptation, ICCV2019

---

> ### Author Response · Authors · 2020-11-25
> **Response to Q3, Q4 and Q5**
>
> `R1-Q3`:  It's somewhat counter-intuitive that the paper required the similarity between the anchor and its augmented version to be closer than the similarity between the anchor and its nearest neighbor (Eq. (9)). A situation may exist: the augmentation is very heavy while the nearest neighbor shows very similar visual features with the anchor. The author has shown the effectiveness of Eq. (9) in Table 2, namely "FR". However, it's not convincing enough, I recommend to evaluate "ENC + SC + SP (NC-SP)", i.e. removing "FR" from the whole model.
>
> `R1-A3`:   Thanks for your constructive comments. We add "ENC-SC- SP" removing “FR” in ablation table 2. We can see that “ENC-SC-SP” achieves comparable performance to “ENC-SC-FR” on office-31 but with minor performance differences for each task. Compared to our ENC-SC-FR-SP (NC-SP), we could see that FR indeed plays an important role in boosting performance. We will elaborate on the possible reason as follows:
> To summarize, there are two situations for the 1st neighbor sample:
> 1) When the first neighbor samples are negative but visually similar to the anchors, introducing this feature ranking regularization benefits the feature representation learning by ensuring the positive samples (self-augmentation) rank higher than the negative samples (the first neighbors) in feature space. This generally makes sense.
> 2) When the first neighbor samples are positive and visually similar to the anchors as you say, it can also enforce an inductive bias on model training to rank the positive samples with pre-defined variations (such as cropping, grayscale, and color distortion) higher than the positive samples with unknown variations (the first neighbors). To make sure that the self-augmentation is not too far from the first neighbors in this case, we should avoid a large triplet margin in Eqn. 9. Note that we add hyperparameter analysis in section 4.5 and Figure 4(b) for the triplet margin.
>
> `R1-Q4`: The paper claimed its state-of-the-art performances on various benchmarks. But it missed some previous methods, e.g. [F]. The paper achieved even inferior performance than [F] on the VisDA17 benchmark, i.e. 86.2 in this paper v.s. 87.2 in [F].
>
> `R1-A4`:  Thanks for pointing out this insightful paper. We agree that our method is 1% behind [F] on VisDA17 benchmark. We also add more comparisons to [F] on office31 and ImageCLEF-DA in table 4 and table 5. We can find that our method is 0.4% better on Office31 and 1.9% better on ImageCLEF-DA than [F].  In general, our method is much simpler than [F] in terms of optimization where they perform alternative optimization between updating the target label hypothesis through k-mean clustering and adapting feature representations through back-propagation.
>
> More importantly, we acknowledge that their inter-class domain discrepancy is orthogonal to our method. We believe that combining with [F] could potentially push our performance to a higher number.
>
> `R1-Q5`:  As I mentioned in weakness-3, it's better to use "minus" rather than "plus" in ablation studies
>
> `R1-A5`:  Thank you for pointing this out. We will correct this in the final version.
>
> `R1-Q6`:  More hyper-parameter analysis would help improve paper quality...
>
> `R1-A6`:  Thanks for your constructive suggestions.  We add the hyper-parameter sensitivity analysis on lambda and triplet margin in section 4.5. Please refer to the text and figure 4 for more details. Overall, our method is robust to the local changes of lambda, and setting the triplet margin too large will harm the performance.
>
> [F] Contrastive Adaptation Network for Unsupervised Domain Adaptation. CVPR 2019.

---

### Official Review · AnonReviewer4 · 2020-10-27
**In this paper, a simple but effective method to impose Neighbor Class Consistency on target features and to learn the discriminative features is proposed.**

**Rating:** 6
**Confidence:** 3

**Review:**

Based on the observation that the target features from source pre-trained model are highly intrinsic discriminative and have a high probability of sharing the same label with their neighbors, a simple but effective method to impose Neighbor Class Consistency on target features is proposed. The experimental results are promising.
[1] Compared with VNC, the improvement of ENC is very limited. Whether the authors have repeated the experiments in several times. It is better to show the variance of the results in Table 2, which may be more suitable to demonstrate the effectiveness of ENC.
[2] In Eq.(9), it forces that the anchor samples should be closed to the neighbor samples by comparing with the distance between the anchor sample and the augmented sample. What is the advantage of such a strategy compared with triplet loss. It is also better to show the comparison in the experiments.
[3] For the neighbors, when k is equal to 1, the proposed method achieved the best performance. Hence, the neighbors with the minimum distance could not guarantee that they share the same labels. How does the proposed method estimate the confidence.

---

> ### Author Response · Authors · 2020-11-25
> **Response to Q1, Q2 and Q3**
>
> `R4-Q1`:  Compared with VNC, the improvement of ENC is very limited. Whether the authors have repeated the experiments several times. It is better to show the variance of the results in Table 2, which may be more suitable to demonstrate the effectiveness of ENC.
>
> `R4-A1`: Thanks for your suggestion. We include the variance of the results in Table 2 in the final version. As you can see from table 2,  ENC consistently outperforms VNC on all 6 tasks with very small variance. It is also worth mentioning that ENC is more robust to neighborhood size K than VNC from figure 3(b), and it can alleviate the negative effect from the false neighbor supervision by assigning less loss weight to them.
>
> `R4-Q2`:  In Eq.(9), it forces that the anchor samples should be closed to the neighbor samples by comparing with the distance between the anchor sample and the augmented sample. What is the advantage of such a strategy compared with triplet loss? It is also better to show the comparison in the experiments.
>
> `R4-A2`: Sorry for causing the confusion. In Eq. 9, we enforce the anchor samples to be closer to their augmented samples than their first neighbors in features space. Specifically, we use triplet loss to introduce this feature ranking constraint between anchors, augmented samples, and anchors’ first neighbors.
>
> For the advantage of such a strategy,
> 1)  When the first neighbors given target anchors are negative  (do not share the same label with anchors),  introducing this feature ranking regularization benefits the feature representation learning by ensuring the positive samples (self augmentation) rank higher than the negative samples in feature space.
>
> 2)  When the first neighbors given target anchors are positive,  it can also enforce a inductive bias on model training to rank the positive samples (self augmentation) with pre-defined variations (such as cropping, grayscale, and color distortion) higher than the positive samples with unknown variations (the first neighbors).
>
> From table 2, we could observe that adding FR improves performance by ~0.5 % on office31. We also add hyper-parameter sensitivity analysis on triplet Marin in fig 4 and section 4.5.
>
>
> `R4-Q3`:  For the neighbors, when k is equal to 1, the proposed method achieved the best performance. Hence, the neighbors with the minimum distance could not guarantee that they share the same labels. How does the proposed method estimate confidence?
>
> `R4-A3`: Thanks for your question. As you notice, the neighbors with the minimum distance in feature space could not guarantee that they share the same labels in the source pre-trained model. The main reason is that the original classifier is largely biased towards the source data.  Based on this motivation, our main contribution is to regularize the classifier less biased by making the class predictions of target neighbors being consistent.
>
> For confidence estimation, our ENC (Eq 6,7) introduces an entropy-based weighting scheme (EW) to assign different loss weight on different neighbor pairs given an anchor. This loss weight can be considered as a measure of confidence estimation. After applying our EW, the negative neighbor pairs will be assigned less loss weight because they have low confidence while positive neighbor pairs will be assigned larger loss weight because they have high confidence. Please also refer to the fig 3(c) which visualizes the loss weight for positive and negative pairs.
>
> From fig 3(b), our method with the entropy-based weighting is robust to the size of k. But for simplicity, we set k equals to 1 for all experiments from table 3-5.

---

### Official Review · AnonReviewer2 · 2020-10-27
**A consistent and effective approach making use of neighbor samples**

**Rating:** 5
**Confidence:** 5

**Review:**

This paper tackles Unsupervised Domain Adaptation. The authors focus on the intrinsic discriminative feature for target samples. The proposed method, Neighborhood Class Consistency among target samples and augmented ones, is proposed as a set of multiple losses to calculate the consistency from several aspects. The experimental results show that the proposed method achieves state-of-the-art performance using the same backbone network.

**Pros**
- The consistent approach making use of neighborhood structure in the target domain is well motivated and easy to understand.
- The proposed method achieves SoTA performance on several benchmark datasets.

**Cons**
- There is a related paper by Gu et al. (2020), although it is not referred to in this paper. Gu et al. also focus on a pseudo labeling approach and discriminative distributions in the target domain. Additionally, the experimental results show that the performance is very close to that in this paper.  Although the detailed method is different, the authors should have compared them qualitatively and quantitatively.

Gu et al., Spherical Space Domain Adaptation with Robust Pseudo-label Loss. In CVPR, 2020.

- Section 4.2 says that the authors use different values for the hyperparameter, such as $\lambda$, for each dataset. How do they tune them? Since the method is for UDA, there cannot be a validation set in the target domain. Therefore, different values of $\lambda$ seem unnatural. Alternatively, the authors should report the performance with different hyperparameter values not only for the number of neighbor samples in Fig. 3 (b) but also for $\lambda$ and triplet margin $m$.

**Minor comments**
- In the main text, there are some points the authors should insert a white space before a bracket and a citation.
- In the last paragraph of Section 1, an abbreviation "NC" is used without an explanation. What does it stand for?
- How often do we need to update the memory bank $\mathbb{V}_t$ while a model is trained? Frequent updates of the memory would be time-consuming.
-The format of References is messy. Some papers, such as CyCADA, should not be arXiv preprints but published ones. Some proceeding names are capitalized while the others are not. Some conference names are abbreviated while the others are not.
-The blank Appendix section should be deleted.

**Overall rating**

Although there should be more descriptions about the existing method and experimental results, the reviewer is leaning toward acceptance. The rating can be upgraded if the authors could solve the cons above.

**Additional comment after rebuttal**

Unfortunately, the reviewer would like to downgrade my first score. The remaining concerns are about R2-A1 and R2-A2.

In R2-A1, the authors add some discussions about experimental results and explanations about Gu et al. (2020). The second discussion about generating pseudo labels would be the most considerable theoretical difference between the proposed method and Gu et al. (2020). The authors state that the proposed method addresses the problem by incorrect pseudo labeling of Gu et al. (2020); however, they fail to show quantitative nor qualitative discussions that support the statement. The third discussion seems just showing the proposed method achieves similar performance to that of Gu et al. (2020). As discussed in the fourth comment, the authors could add another result showing the proposed method is complementary to Gu et al. (2020).

In R2-A2, the authors honestly state that the hyperparameters are tuned according to the test data evaluation. However, it should be avoided to evaluate unsupervised domain adaptation methods since there are no labeled data in the target domain in a real setting. Moreover, Figure 4 (a) shows that the proposed method is sensitive to lambda on Office31 dataset. When lambda=0.5 on all dataset, the proposed method achieves SOTA on VisDA as shown in Table 5, but seems to fail on Office31.

---

> ### Author Response · Authors · 2020-11-25
> **Response to Question 1**
>
> `R2-Q1`:There is a related paper by Gu et al. (2020) [4] also focus on a pseudo labeling approach and discriminative distributions in the target domain. Additionally, the experimental results show that the performance is very close to that in this paper. Although the detailed method is different, the authors should have compared them qualitatively and quantitatively.
>
> `R2-A1`:Thanks for pointing out this insightful paper and we will definitely cite it in the revised version. We would like to kindly elaborate the distinct differences between Gu et al. (2020) and our NC-SP as follows.
>
> 1) Gu et al. (2020) projects images to a spherical feature space where domain alignment techniques are performed, while we focus on enforcing consistency constraints in a standard feature space.
>
> 2) One of the domain alignment techniques in Gu et al. (2020) is based on pseudo labelling in the spherical space, which bears some similarity with NC-SP. But Gu et al. (2020) generates hard pseudo labels (one-hot vectors obtained by the argmax operator), where the incorrect pseudo ones may bring negative impact for the performance. This is exactly the problem which NC-SP aims to address. NC-SP utilizes  high-quality pairwise neighbor supervision. No hard pseudo labels are generated in the process.
>
> 3) It is worth noting that Gu et al. (2020) combine several loss terms from prior literature such as conditional entropy minimization[1],  adversarial domain classification term [2], Semantic matching loss[3]. We list the comparison of Gu et al. (2020) and NC-SP as belows. We can see that NC-SP is close to Gu et al. (2020) in office-31 while better in ImageCLEF-DA. It is also worth mentioning that our NC-SP can achieve much better performance than “Gu et al. (2020) without using other external loss terms such as [1][3]”.
>
> |  Method              |  Office-31  | ImageCLEF-DA |
> |-|-|-|
> | Gu et al. (2020)                  |91.1            |90.5|
> | Gu et al. (2020) wo [3]     | 90.2            | 90.1 |
> | Gu et al. (2020) wo [1][3]| 89.2            | 89.4 |
> | NC-SP                 | 91.0            |90.9|
>
>
> 4) More importantly, our proposed method is orthogonal with the spherical alignment techniques proposed in Gu et al. (2020). We can apply the proposed consistent constraints in the spherical space to further mitigate the domain shifts, along with other techniques proposed in Gu et al. (2020) such as [1][3], which very likely contributes to even better performance.
>
> [1]Ruijia Xu, Guanbin Li, Jihan Yang, and Liang Lin. Larger norm more transferable: An adaptive feature norm approach for unsupervised domain adaptation. In ICCV, 2019.
>
> [2] Yaroslav Ganin, Evgeniya Ustinova, Hana Ajakan, Pascal Germain, Hugo Larochelle, Franc¸ois Laviolette, Mario Marchand, and Victor Lempitsky. Domain-adversarial training of neural networks. JMLR, 17
>
> [3]  Shaoan Xie, Zibin Zheng, Liang Chen, and Chuan Chen. Learning semantic representations for unsupervised domain adaptation. In ICML, 2018
>
> [4] Gu et al., Spherical Space Domain Adaptation with Robust Pseudo-label Loss. In CVPR, 2020.

---

> ### Author Response · Authors · 2020-11-25
> **Response to Question 2, 3 and 4**
>
> `R2-Q2`:  Section 4.2 says that the authors use different values for the hyperparameter, such as λ, for each dataset. How do they tune them? Since the method is for UDA, there cannot be a validation set in the target domain. Therefore, different values of λ seem unnatural. Alternatively, the authors should report the performance with different hyperparameter values not only for the number of neighbor samples in Fig. 3 (b) but also for  λ and triplet margin m
>
> `R2-A2`:  Thanks for your constructive suggestions.  We add the hyper-parameter sensitivity analysis on lambda and triplet margin in section 4.5. Please refer to the text and figure 4 for more details. Overall, our method is robust to the local changes of lambda around 0.1, and setting the triplet margin too large such as 1 will harm the performance.
>
> As you notice, we choose different λ for office31 and VisDA17. For tuning those λ, we are referring to the performance of the test dataset. Similarly to figure 4, we tuned the λ from  { 0.1, 0.25, 0.5, 0.75} on VisDA17 and found that λ=0.5 achieved the best performance with 86.2 %. Note that the performance of λ=0.1 is 85.7 %. As the performance does not change a lot for those two choices of λ,  we are safe to say that our method is still robust to the regional changes of hyper-parameter λ in VisDA17.
>
> `R2-Q3`: How often do we need to update the memory bank Vt while a model is trained? Frequent updates of the memory would be time-consuming.
>
> `R2-A3`:Thanks for pointing this out. We update the memory bank at every epoch. But we would like to kindly argue that it will not introduce extra computational cost but more time-efficient.
>
> 1) Before the training starts, we feed forward the entire dataset to the model, get the target features for each target id and store them in the memory bank Vt. Then we apply kNN on the memory bank Vt for neighborhood discovery in Section 3.3.1. Thus, we can associate each target id with its k nearest neighbors.
>
> 2) When the training starts, we optimize our model in a batch-wise manner. Given the current batch, we extract the target features and retrieve their kNN features from the memory bank for computing loss. Note that we retrieve the kNN features directly from the memory bank instead of extracting them again from raw images by enlarging the batch size. Thus, using a memory bank will be k times faster than without it. After finishing the current batch update, we do not discard the target features of the current batch but use it to update (technically replace) the target features of the same id inside the memory bank. This process will not introduce any computational burden.
>
> Thanks again for pointing this out. We will include this detail in the final submission.
>
> `R2-Q4`:Authors should insert a white space before a bracket and a citation; The format of References is messy; an abbreviation "NC" is used without an explanation; The blank Appendix section should be deleted.
>
> `R2-A4`:Thank you for pointing this out. We will correct this and make the references correct and consistent in the final version.
>
> [1]Ruijia Xu, Guanbin Li, Jihan Yang, and Liang Lin. Larger norm more transferable: An adaptive feature norm approach for unsupervised domain adaptation. In ICCV, 2019.
>
> [2] Yaroslav Ganin, Evgeniya Ustinova, Hana Ajakan, Pascal Germain, Hugo Larochelle, Franc¸ois Laviolette, Mario Marchand, and Victor Lempitsky. Domain-adversarial training of neural networks. JMLR, 17
>
> [3]  Shaoan Xie, Zibin Zheng, Liang Chen, and Chuan Chen. Learning semantic representations for unsupervised domain adaptation. In ICML, 2018

---

### Official Review · AnonReviewer3 · 2020-10-28
**The idea of neighbor consistency heavily lies on the assumption that target features from source pre-trained model are highly intrinsic discriminative. It is a controversial assumption.**

**Rating:** 5
**Confidence:** 3

**Review:**

This paper addresses the unsupervised domain adaption (UDA) problem. Particularly, the paper proposes to impose neighbor class consistency on target features to preserve intrinsic discriminative nature of target data and presents an entropy-based weighting scheme to improve robustness against the potential noisy neighbor supervision. The motivation of the paper is clear and the method is well presented. Extensive experiments show the effectiveness of the proposed method. However, the paper suffers some problems, such as
1. The idea of neighborhood consistency in UDA is not very significant. It lies on the assumption that target features from source pre-trained model are highly intrinsic discriminative. However, the distribution discrepancy between source domain and target domain may be very large, such as unsupervised cross-dataset person re-identification. It is not sure whether such assumption/observation is still satisfied.
2. From Table 3-5, NC-SP outperforms the state-of-the-art methods with only a small margin. From Table 3, SC, FR , SP make a great contribution to the performance. It is suggested to add some comparisons to Table 3-5 that only the proposed VNC or ENC is used without other tricks.
3. It is suggested to evaluate the influence of hyper-parameter \lambda in Eq. (10) and (11)
4.  It is suggested to evaluate neighbor size K on more datasets, in order validate the association of K with target dataset and number of classes.

---

> ### Author Response · Authors · 2020-11-25
> **Response to Question 1 and 2**
>
> `R3-Q1`: The idea of neighborhood consistency in UDA is not very significant. It lies on the assumption that target features from source pre-trained models are highly intrinsic discriminative...
>
> `R3-A1`: We sincerely appreciate your insightful comments.  We agree that the domain discrepancy has a large impact on the performance of our method as you can see that the performance on VisDA17 dataset (in Table 5), where the domain discrepancy is very large,  is much lower than the Office31 dataset  (in Table 3). But we want to kindly argue from the following three points.
> 1) **Our method is more robust to domain discrepancy w.r.t performance**:The existing methods such as adversarial-based domain adaptation methods or self-training based methods also suffer from performance drop when domain discrepancy is large. e.g in VisDa17. Compared to those baselines in the VisDa dataset, our method still outperforms them by a significant margin. It demonstrates that our method is robust and can achieve consistent better performance in datasets with different levels of domain discrepancy than the baselines.
> 2) **Existing Self-training methods and ours are based on the similar discriminative assumption of source model while ours relies on a looser assumption which is easier to achieve**: Specifically, our method shares similar motivation with Self-training methods in terms of leveraging the Pseudo label for unlabeled target data. In comparison, we utilize the pairwise pseudo label while they use class-wise pseudo labels. As source and target data share the label space in unsupervised domain adaptation, both two threads of methods follow the similar assumption that the feature representation has explored some degree of discriminativeness with the help of source data pre-training. But compared to the class-wise pseudo label which relies on a hard assumption that the target features are globally discriminative across all categories, our pairwise pseudo label relies on the looser assumption that the target features are locally intrinsic discriminative (target features share the same label with their k nearest neighbors). Therefore, we claim that our method is more robust to domain discrepancy than self-training methods with the looser assumption especially when k is small.  It is also worth noting that the self-training method [1] based on the source pre-trained model also achieves state-of-the-art performance in cross-domain person RE-ID tasks where the discrepancy is large. We think both self-training and our methods are effective in UDA and UDA person re-ID with further pseudo label refinement.
> 3) **We proposed specific components to improve the discriminativeness of initial feature space**:To ensure the intrinsic discriminativeness of feature space, we propose the self-consistency loss pre-training(SP) and feature ranking between self-augmentation and the first neighbor. From Table 2, we can see that our method improves by a large margin with those important components.
>
> `R3-Q2`: NC-SP outperforms state-of-the-art methods with only a small margin. It is suggested to add some comparisons to Table 3-5 that only the proposed VNC or ENC is used without other tricks.
>
> `R3-A2`: Thanks for your constructive advice. We would like to kindly argue that the improvement of our method on Table 3-5 is not trivial in UDA task. Also, it is worth noting that we introduce a simple and new way of utilizing the pseudo label while outperforming the state-of-the-art methods which are generally more complex and usually combine several techniques together. For example, in table 3,  SRDC is based on a clustering method [2] with source sample selections. In table 4, ALP claims that it is orthogonal to existing methods and built on top of another paper “CAN” [3].
>
> Additionally, we would like to add ENC as a comparison in three datasets for your reference in appendix Table 7 and Table 8. Thanks for pointing it out. Note that the ablation table 2 and table 3 both refer to the Office-31 dataset and thus you can compare the ENC result in table 2 directly with other state-of-the-art methods in table 3. We could observe that our method with ENC only achieves the third best performance (exclude NC-SP) in Office-31, top 1 performance in ImageCLEF-DA and the second best performance in VisDA17.  Compared to the self-training based methods (ENT, MRENT, SAFN+Ent), our ENC outperforms them by a large margin without bells and whistles.
>
> (To be continued on the next page.)
>
> [1] Ge, Yixiao, Dapeng Chen, and Hongsheng Li. "Mutual mean-teaching: Pseudo label refinery for unsupervised domain adaptation on person re-identification." ICLR (2020).
>
> [2] K. G. Dizaji, A. Herandi, C. Deng, W. Cai, and H. Huang. Deep clustering via joint convolutional autoencoder embedding and relative entropy minimization. ICCV 2017
>
> [3] G. Kang, L. Jiang, Y. Yang, and A. G. Hauptmann. Contrastive adaptation network for unsupervised domain adaptation, CVPR2019

---

> ### Author Response · Authors · 2020-11-25
> **Response to Question 2 , 3 and 4**
>
> `R3-Q2`: NC-SP outperforms state-of-the-art methods with only a small margin. It is suggested to add some comparisons to Table 3-5 that only the proposed VNC or ENC is used without other tricks.
>
> `R3-A2`: It is also worth mentioning that we think Self class consistency (SC), Feature ranking between self-augmentation and the first neighbor (FR), SP (self-consistency pre-training) are all important components of our method. Specifically, SC is the special case of Neighbor class consistency where the nearest neighbor given an anchor is itself. FR provides a regularization constraint in feature space to ensure the intrinsic discriminativeness.  Similarly, SP could also help result in a better feature representation initialization for more precise neighborhood discovery.
>
> `R3-Q3`: It is suggested to evaluate the influence of hyper-parameter \lambda in Eq. (10) and (11).
>
> `R3-A3`:  Thanks a lot for your constructive suggestion. We add the hyper-parameter sensitivity analysis on lambda and triplet margin in section 4.5. Please refer to the final submission and figure 4 for more details. Overall, our method is robust to the local changes of lambda, and setting the triplet margin too large will harm the performance.
>
> `R3-Q4`: It is suggested to evaluate neighbor size K on more datasets, in order to validate the association of K with the target dataset and number of classes.
>
> `R3-A4`: Thanks for pointing it out. We add an additional figure 5 in appendix section A1 for analyzing neighbor size K on all tasks in Office31 (3 datasets and 6 tasks). We can conclude that our method is very robust to the size of K where k=1 and k=5 have similar performance.
>
>
>
> [1] Ge, Yixiao, Dapeng Chen, and Hongsheng Li. "Mutual mean-teaching: Pseudo label refinery for unsupervised domain adaptation on person re-identification." ICLR (2020).
>
> [2] K. G. Dizaji, A. Herandi, C. Deng, W. Cai, and H. Huang. Deep clustering via joint convolutional autoencoder embedding and relative entropy minimization. ICCV 2017
>
> [3] G. Kang, L. Jiang, Y. Yang, and A. G. Hauptmann. Contrastive adaptation network for unsupervised domain adaptation, CVPR2019

---

### Author Response · Authors · 2020-11-25
**Appreciate your valuable comments and time**

We sincerely appreciate your valuable comments that help us improve the paper.

---

### Decision · Program_Chairs · 2021-01-07
**Final Decision**

**Decision:**

Reject

**Comment:**

This paper proposes a method (via a novel objective) for feature alignment between source and target tasks in an unsupervised domain adaptation scenario.

Pro
- the proposed approach is sensible in many realistic scenarios of distribution shift
- the submission provides an extensive empirical evaluation establishing state of the art results on several benchmark tasks

Con
- there is no thorough discussion of the the underlying assumptions (when should we expect them to hold? for shich types of tasks? which types of shifts? can they generally be reliably tested? from which type of data? unlabeled?)
- one reviewer raised concerns over novelty, which should be more clearly addressed before publication
- two reviewers raised concerns over use of target data for hyper-parameter selection, which seem valid; these should be fixed or clearly explained (and implications of this be discussed) in subsequent versions of this work

I agree with concerns of the reviewers (the last two points), and would therefore not recommend this work for publication in the current state.